# Antioxidant and Sensory Properties of Raw and Cooked Pork Meat Burgers Formulated with Extract from Non-Compliant Green Coffee Beans

**DOI:** 10.3390/foods12061264

**Published:** 2023-03-16

**Authors:** Monica Bergamaschi, Nicoletta Simoncini, Vincenzo Maria Spezzano, Maura Ferri, Annalisa Tassoni

**Affiliations:** 1Dipartimento Carni, Stazione Sperimentale per l’Industria delle Conserve Alimentari, Viale Tanara 31/A, 43121 Parma, Italy; nicoletta.simoncini@ssica.it (N.S.); vincenzomaria.spezzano@studenti.unipr.it (V.M.S.); 2Dipartimento di Scienze degli Alimenti e del Farmaco, Università degli Studi di Parma, Parco delle Scienze 27/A, 43124 Parma, Italy; 3Department of Biological, Geological and Environmental Sciences, Alma Mater Studiorum—University of Bologna, Via Irnerio 42, 40126 Bologna, Italy; maura.ferri@unibo.it (M.F.); annalisa.tassoni2@unibo.it (A.T.)

**Keywords:** green coffee bean extract, natural antioxidants, antioxidant capacity, pork burgers, lipid oxidation

## Abstract

The effects of polyphenol-rich extract obtained from non-compliant defatted green coffee beans (dGCBs) on physicochemical and antioxidant properties, as well as on the sensory profile of vacuum-packed pork burgers stored at 4 °C for 14 days and after cooking were assessed. The dGCB extract obtained by means of supercritical water extraction was analyzed for its polyphenol profile, total phenolic content, radical scavenging, and ferric-reducing antioxidant activities (DPPH and FRAP), Fe^2+^-chelating capacity, and total iron. The most abundant polyphenol component observed in the dGCB extract was chlorogenic acid, and the alkaloid caffeine was also present. This extract showed antioxidant properties. Thereafter, five formulations of pork meat burgers with added NaCl (1%) were prepared; one without the antioxidant (negative control, C) and one with the use of a synthetic antioxidant (0.05% ascorbic acid = positive control, A), while the other three were supplemented with a different amount of dGCB extract (P15 = 0.15%; P30 = 0.30%; P60 = 0.60%). The addition of dGCB extract increased the antioxidant activity of the raw and cooked burgers and reduced the lipid oxidation of the cooked burgers (0.47, 0.21, and 0.20 vs. 1.28 and 0.55 mg MDA eq./Kg, for P15, P30, and P60 vs. C and A, respectively). No negative effects were observed on the meat’s color parameters and its stability during refrigerated storage and after cooking, nor on sensory attributes (color and aroma) for the lowest concentration of coffee extract. The results obtained indicate that 0.15% dGCB extract is a promising alternative to commercial synthetic antioxidants to improve the quality of refrigerated pork burgers.

## 1. Introduction

In recent decades, increasing consumption of meat has been especially related to meat coming from pork [1]. Medium-term projections show that pork will remain the most eaten meat in the European Union over the next ten years [2]. In Italy, the apparent consumption of fresh meat was around 675 thousand tons in 2021, corresponding to 11.3 kg/year per capita, with an increase of 2.9% compared to 2020 [3]. This growth was probably favored by greater availability on the market of ready-to-cook pork products, which better meet the modern needs of the consumer [4]. The main issue in pork meat preparations is the preservation of its quality, because the loss in nutritional and sensory value occurs during shelf life and cooking [5,6,7,8], as well as oxidative degradations related to harmful effects on health [9,10].

Lipid and color oxidation are the main non-microbial causes of quality loss in ready-to-cook pork products, such as burgers. Several studies have confirmed the existence of a strong relationship between lipid and color oxidation [11,12]. Lipid oxidation results in a rancid odor, off-flavor development, the loss of nutrients, reduced shelf life, and the generation of some toxic compounds that compromise consumer health [13,14,15]. Meat discoloration is caused by the oxidation of the central iron atom within the heme group of the myoglobin, which changes from red oxymioglobin to brownish metmyoglobin [16]; it is muscle specific and encouraged by high temperatures, low pH values, low partial oxygen pressures, and the presence of non-heme iron [17,18]. 

In the meat industry, it is common to add synthetic antioxidants to meat preparations to prevent lipid oxidation and color fading. However, in recent years, there has been increased interest in replacing these synthetic antioxidants with natural substances, as demanded by the food market [19,20]. Recent studies have reported that natural antioxidants from plant matrices, waste, and by-products represent a promising alternative to synthetic substances as they contain high amounts of polyphenols and other bioactive compounds that are effective in preserving the red color of meat and in reducing the formation of dangerous products of oxidation [21,22]. High antioxidant activities and biological properties are also reported for coffee polyphenols [23,24,25,26]. In particular, green coffee beans are a rich source of polyphenols; they mainly contain hydroxycinnamic acids, such as caffeic and ferulic acid, as well as their esters with quinic acid, which are referred to as chlorogenic acids [27]. 

In the coffee supply chain, about 20% of green coffee beans are discarded due to non-compliance compromising the final quality of the coffee. The main defects are related to the presence of physical damage (e.g., broken and insect-damaged beans), irregularity in shape and size, and anomalous colors, odors, and textures [28]. These waste products are a promising source of ingredients that are rich in antioxidant compounds and might be used in food. The addition of plant extracts into the meat matrix can be difficult if their color and flavor are not suitable for meat and if their solubility is insufficient, generating unpleasant spots in the final product [29,30]. Some authors investigated the effectiveness of the subcritical water extraction (SWE) method for the recovery of water-soluble phenolic compounds from medicinal plants and coffee by-products [31,32], as well as the possibility of producing plant extracts that can be used as natural antioxidant ingredients in pharmaceuticals, cosmetics, and food [33,34,35,36]. During the last decade, some studies have reported increased oxidative stability and improved functional properties of food products prepared with extracts from green coffee beans, which are rich in polyphenols [27,37,38], but there are not results on their use as an ingredient in pork meat. 

The aim of this work is to investigate the physicochemical properties, the lipid and color stability, and the sensory characteristics of raw pork burgers during refrigerated storage and after cooking, prepared by replacing synthetic ascorbic acid with a natural polyphenol-rich extract obtained from non-compliant green coffee beans.

## 2. Materials and Methods

### 2.1. Characterization of the Defatted Green Coffee Bean (dGCB) Extract

The dGCB extract used as a natural antioxidant in pork meat burgers was supplied by the Bio Base Europe Pilot Plant (Gent, Belgium). The technical-grade extract was a fine brownish powder, which was obtained by freeze-drying the polyphenol-rich aqueous solution that was extracted from non-compliant *C. arabica* coffee green beans through a defatting step, followed by a sub-critical water extraction (SWE) process, developed at a laboratory scale by CELABOR (Harve, Belgium). To administer adequate doses of the extract to the burgers, the extract was analyzed for its total phenolic content (TPC), polyphenol profile, antioxidant activities (DPPH and FRAP assays), Fe^2+^-chelating capacity, and total iron content. 

The dGCB extract was characterized for total phenolic content (TPC) according to the method described by Banerjee et al. [39], which is based on the Folin–Ciocalteau assay. TPC was calculated using a calibration curve prepared with gallic acid (Sigma-Aldrich, St. Louis, MO, USA) and expressed as equivalent to g of gallic acid per kg of the extract (g GA eq.)/kg.

Aliquots (50 mg/mL of Milli-Q^®^ water) of dGCB extract were analyzed for the identification and quantification of specific polyphenols and caffeine by means of HPLC-DAD. Chromatographic analyses were performed using a high-performance liquid chromatography system (column Gemini C18, 5 μm particles 250 mm × 4.6 mm, pre-column Security Guard Ea, Phenomenex, Torrence CA, USA) equipped with an online diode array detector (MD-2010, Plus, Jasco Instruments, Großumstad, Germany), as described by Ferri et al. [40]. The HPLC-DAD separation procedure allowed for the simultaneous analysis of caffeine and chlorogenic acid. For each sample, two chromatograms obtained at different wavelengths (270 and 323 nm) were analyzed to determine the concentration of the single compounds, depending on their maximum absorbance. The analysis was performed in triplicate, and the results are expressed as mg/g of the extract.

The antioxidant capacity of the dGCB extract was assayed as its 1,1-diphenyl-2-picrylhydrazyl (DPPH) free-radical-scavenging ability and ferric-reducing antioxidant power (FRAP), following the method described by Serpen et al. [41]. Proper dilution with cellulose was undertaken, as proposed by Saccani et al. [42], due to the high antioxidant activity. The antioxidant capacity was expressed as mmol Trolox equivalents per kg of the sample (mmol Trolox eq.)/kg, using calibration curves with 6-hydroxy-2,5,7,8-tetramethylchroman-2 carboxylic acid (Trolox) in the range 0–30 µM.

Fe^2+^-chelating activity was assessed by measuring the formation of the Fe^2+^-ferrozine complex according to the method described by Liu et al. [43] with some modifications. One gram of the extract was added to 20 mL of acidic methanol–water (50:50, *v*/*v*, pH 2). The mixture was sonicated in an ice bath for 30 min, and then shaken at room temperature for 30 min. After centrifugation (1800× *g*) at 4 °C for 20 min, the supernatant was filtered and stored, while the pellet was extracted again with 20 mL of acetone: water solution (70:30, *v*/*v*), sonicated, shaken, centrifuged, and filtered as described above. The methanolic and acetonic extracts were mixed and evaporated to dryness under a vacuum at 40 °C. The dried extract was recovered in 10 mL deionized water and stored at −20 °C until analysis. An aliquot of the water solution (0.5 mL) was mixed with 2 mL of sodium acetate buffer (0.1 M, pH 4.9) and 50 μL of iron (II) chloride (2 mM). After 30 min of incubation at room temperature, 0.2 mL ferrozine (5 mM) was added. After 30 min, the absorbance was measured at 562 nm. Distilled water was used as the control. The percentage of inhibition of Fe^2+^–ferrozine complex formation was calculated as [(A0 − A1)/A1] × 100, where A0 was the absorbance of the control and A1 was the absorbance of the extracts. The IC50 value refers to the effective extract concentration (mg/mL), at which 50% of the ferrous (Fe^2+^) ions were chelated; this was calculated by linear regression analysis of the data. 

The concentration of total iron in the dGCB extract was measured by nitric acid digestion [44] using an UltraWAVE SRC Microwave Digestion System (Milestone Inc., Shelton, CT, USA) to dissolve the organic material and make heavy metals available for measurement with flame atomic absorption spectroscopy (FAAS). Aliquots (0.5 g) of samples were combined with 4 mL of HNO3 Ultrex II Ultrapure Reagent (J.T.Baker™, Thermo Fisher Scientific Inc., Waltham, MA, USA) in a glass test tube with a Teflon cap. The samples were loaded into the UltraWAVE, pressurized to 50 bars, and heated to 250 °C for 20 min. After cooling, samples were quantitatively transferred to 25 mL volumetric flasks and filled to the mark with Milli-Q^®^ water. Total iron was quantified by FAAS (Varian AA2- 40, Mulgrave, Australia), operating in absorption mode, with hollow cathode lamps (248.3 nm) and an air–acetylene flame. An iron standard curve with a concentration range of 0.2–4.6 mg/L was obtained daily by the appropriate dilution of the stock solution containing 1000 mg/L (Fluka, Buchs, Germany). Accuracy was confirmed with the certified reference material, bovine muscle powder (ERM^®^-BB184). The results are expressed as μg of total iron per gram of extract. 

### 2.2. Pork Burger Manufacture and Storage

Fresh loins and neck muscles were purchased in a local slaughterhouse and stored at 2 ± 1 °C overnight. Then, 60% loin and 40% neck muscles were ground using a 4.5 mm Ø grinding plate, and the meat samples were collected for proximate composition (moisture, crude protein, fat, and ash), which was determined following the AOAC’s official methods [45]. Salt (NaCl, 1%) was added to the ground meats, which were then divided into five 4 kg batches. The following five formulations (F) were prepared: a formulation with only meat and salt (negative control, C); one with meat, salt and 0.05% ascorbic acid (positive control, A); and three formulations with meat, salt, and either 0.15% (P15), 0.30% (P30), or 0.60% (P60) dGCB extract. In total, 40 burgers (100 g in weight, 10 cm in diameter, and 1 cm thick) per formulation were made using a burger hand-forming press. The burgers were left in a blast chiller at −40 °C for 30 min; then, they were individually vacuum-packed with skin-pack technology (CRYOVAC^®^ Darfresh^®^, Sealed Air Corporation, Charlotte, NC, USA) and, finally, they were stored at 4 ± 1 °C in a thermostat. Three raw and three cooked burgers per formulation and per replicate were analyzed at 0, 7, and 14 days of storage (storage time, ST; T0, T7, and T14). The burgers were displayed under a pink fluorescent lamp (Osram Natura T8, 18 W, 750 lm) in a refrigerated counter for 24 h before sampling in order to mimic supermarket retail-display conditions. Cooking was carried out in an oven preheated to 180 °C until the internal temperature of the burgers reached 70 °C. The temperature at the core was measured using an infixion probe connected to the E-Val flex Thermal System (Ellab, Hillerød, Denmark). The whole experiment was repeated twice on different days, resulting in two independent replicates per formulation.

### 2.3. Burger Sampling and Analysis

Prior to the storage period (T0) and after 7 and 14 days of storage (T7 and T14), three raw and three cooked burgers per formulation and per replicate were finely chopped with a knife once the instrumental color measurements were taken. The pH and moisture were assessed for each of the samples immediately after mincing. Aliquots (30 g) of one finely chopped burger per formulation and per replicate were wrapped in aluminum foil, vacuum packaged, stored at −20 °C, and analyzed after a month for lipid peroxidation (TBARS test), total phenolic content, antioxidant capacities (DPPH and FRAP tests), and heme and non-heme iron content. The other three fresh and three cooked burgers were used for sensory analysis.

#### 2.3.1. Total Phenolic Content (TPC) and Antioxidant Activities

Total phenolic content (TPC) in fresh and cooked burgers was determined using the Folin–Ciocalteu assay method described by Banerjee et al. [39], with some modifications. Five grams of each sample were homogenized (IKA Ultra Turrax, Staufen, Germany) with 25 mL of 50% methanol 0.02 N HCl solution, kept for 30 min in an ultrasonic system in an ice-water bath, and stored overnight at 4 ± 1 °C. After centrifugation (1800× *g*) at 4 °C for 20 min, the supernatant was filtered and stored, while the pellet was extracted again with 25 mL of acetone: water solution (70:30, *v*/*v*), sonicated, shaken, stored for 4 h at 4 ± 1 °C, then centrifuged and filtered as previously described. Total phenolic content was calculated by adding the phenolic content of the methanolic and acetonic solutions and determined using a calibration curve prepared with each extraction solvent. The results are expressed as g of gallic acid equivalents per kg of sample (g GA eq./kg).

DPPH and FRAP activities were assessed in freeze-dried samples according to the procedure reported by Serpen et al. [41]. The results are expressed as mmol of Trolox equivalents per kg of dried weight sample (mmol Trolox eq./kg_dw_). 

#### 2.3.2. Thiobarbituric Acid Reactive Substances (TBARS)

The lipid oxidation of raw and cooked burgers was evaluated spectrophotometrically (Spectrophotometer V-730, Jasco, Easton, MD, USA) by measuring the thiobarbituric acid reactive substances (TBARS) according to the method described by Witte et al. [46]. The absorbance was read at 532 nm and corrected by subtracting the absorbance at 600 nm for sample turbidity. The results are expressed as mg of malondialdehyde equivalents per kg of sample (mg MDA eq./kg) using a 1,1,3,3-tetramethoxypropane standard curve.

#### 2.3.3. Heme Iron (HI) and Non-Heme Iron (NHI)

Heme iron was assessed according to the method described by Pretorius et al. [5], with some modifications. Samples of approximately 5 g were weighed in a 50 mL polypropylene tube and homogenized with 25 mL of the extraction solution using an Ultra Turrax homogenizer (T25, IKA, Königswinter, Germany) at 15,000 rpm for 1 min. The amount of added extraction solution (ratio of acetone to water to concentrated hydrochloric acid, 40:9:1) included the contribution of the sample moisture. The homogenized sample was stored at 2 °C in the dark for 60 min, then it was centrifuged for 10 min at 10,000× *g* at 4 °C. Thereafter, the supernatant was filtered through a glass microfiber filter (GF/A, Whatman, Little Chalfont, UK) and the absorbance was measured at 640 nm against a blank reagent. The absorbance was multiplied by 680 and by the dilution factor to give the concentration as mg of hematin per kg of sample. The iron in hematin was calculated by multiplying by 0.0882 (the ratio between the atomic weight of iron and the molecular weight of hematin) and dividing it by 10 in order to express the results as mg of heme iron per 100 g of sample.

Non-heme iron (NHI) was spectrophotometrically assessed according to the method described by Ahn at al. [47], with some modifications. Five grams of the sample were weighed into 50 mL test tubes and homogenized (IKA Ultra Turrax) with 12 mL of 0.1 M citrate-phosphate buffer (pH 5.5) for 30 s. Then, 4 mL of 2% ascorbic acid in 0.2 N HCl were added, and the mixture was incubated at room temperature for 15 min. The proteins were precipitated with 8 mL of 11.3% trichloroacetic acid (*w*/*v*); then, the mixture was centrifuged at 3000× *g* for 10 min and the supernatant was filtered using a paper filter (Whatman n° 1). A total of 2 mL of clear supernatant were mixed with 0.8 mL of 10% ammonium acetate (*w*/*v*) and 0.2 mL of ferrozine color reagent into a disposable cuvette (Kartel, 1937). The absorbance was read at 562 nm after 5 min. The non-heme iron concentration was calculated using a Fe(II) standard curve and the results are expressed as mg of non-heme iron per 100 g of sample. 

#### 2.3.4. Instrumental Color and pH

Instrumental color parameters were measured on the burger surface (five measurements) before and after cooking, avoiding areas with a lot of fat. A CM 700-d spectrophotometer (Minolta, Osaka, Japan) was used, and the following operating conditions were applied: illuminant D65; observation angle 10°; measuring hole 1 cm; and the specular component excluded. The color coordinates L* (lightness), a* (redness), and b* (yellowness) were obtained using the CIELab system space, and Chroma [√(a*2 + b*2)] and Hue [tan-1 (b*/a*)] were calculated.

The pH was measured directly on raw and cooked burgers at room temperature using a pH meter (Five Easy F20, Mettler Toledo, Greifensee, Switzerland) equipped with a glass electrode (XS Sensor 2-pore T DHS, Sensor Instruments, Thurmansbang, Germany), after calibration with pH 7.0 ± 0.02 and 4.0 ± 0.02 buffer solutions. The measurements were performed in three different positions of each sample.

#### 2.3.5. Cooking Loss Percentage (CL%)

The burgers were weighed before and after cooking when they reached room temperature, according to the method described by Bellucci et al. [48]. The cooking loss percentage (CL%) was calculated according to the following formula: CL (%) = [(W_bc_ − W_ac_)/W_bc_] × 100, where W_bc_ and W_ac_ are the weights (g) of the burger before and after cooking, respectively. 

#### 2.3.6. Sensory Analysis

Raw and cooked burgers were evaluated by quantitative descriptive analysis [49], using a panel made up of eight staff members of the Meat Department of SSICA. The members were selected and trained according to the voluntary ISO 8586 standards. Informed consent was obtained from all panel members involved in this study. In each session, corresponding to each sampling time (0, 7, and 14 days), the panelists evaluated the raw and then the cooked burgers; the samples were presented according to a balanced design aimed at minimizing the effect of order and carry over [50]. The following attributes were considered: appearance; color intensity (red for raw meat or brown for cooked meat) and homogeneity; typical aroma intensity (defined as the intensity of the characteristic aroma of the raw or cooked meat); the presence of off-odors; rancid odors; and the overall acceptability. The cooked burgers were not tasted. Each attribute was scored on a non-structured 0–9 intensity scale (0 = not detected, 9 = maximum perception of the attribute).

### 2.4. Data Analysis

Data related to the chemical, technological, and sensory variables of raw and cooked burgers were analyzed using the general linear model (GLM) feature of SPSS statistic software (version 22.0.0, IBM, Somers, NY, USA). The formulation (F) and the storage time (ST) were assigned as fixed factors and their interaction (F × ST) was included; the manufacturing repetition was a random effect. The linear model used was as follows:$$Y_{ijk}=\mu +\alpha_{i}+\beta_{j}+{\left(\alpha \beta \right)}_{ij}+\varepsilon_{ijk}$$
where *Y_ijk_* is the dependent variable of the *k*th observation, *μ* is the overall mean, *α_i_* is the effect of the formulations (F, *i* = C, A, P15, P30, P60), *β_j_* is the effect of the storage days (ST, *j* = T0, T7, T14), (*αβ*)*_ij_* is the effect of the interaction between F and ST (F × ST), and *ε_ijk_* is the residual random error associated with the *k*th observation. Tukey’s post-hoc test was used to determine the significant differences between means of groups for the treatment F and ST (*p* < 0.05), when there was a significant source of variation (*p* < 0.05). As the interaction F × ST was not significant (*p* > 0.05) for all dependent variables, it was not reported. The results were reported as estimated marginal means and the variability as standard error of the mean (SEM) for fixed effects F and ST.

## 3. Results

### 3.1. Chemical Profiling and Antioxidant Activities of the dGCB Extract

The dGCB extract used as a natural antioxidant in this study was characterized in order to establish the appropriate range of concentration to be added to the burgers. We determined its total phenolic compounds, polyphenol components, reducing capacity according to the FRAP method, capacity to scavenge DPPH free radicals, and Fe^2+^-chelating activity (Table 1).

The applied SWE method was effective in extracting a high concentration of water-soluble phenolic compounds. Hall et al. [51] reported values of total phenolic compounds for C. arabica green coffee beans of different origins in the range of 4–8%. Furthermore, chlorogenic acid and its derivatives comprise the majority of these compounds. The TPC value measured in 1 g of dGCB extract was similar to the amount detected in freeze-dried extracts of roasted Arabica coffee beans and in a serving of Turkish coffee [52,53]. The most abundant polyphenol component observed in the dGCB extract was chlorogenic acid. The alkaloid caffeine was also present in large amounts (Table 1). The concentration of chlorogenic acid was similar to the amount detected in freeze-dried extracts obtained in other studies by directly boiling a blend of ground green coffee beans and water, while a greater concentration of caffeine was measured [24,25,27]. Other authors have reported a high positive relationship between the content levels of TPC, chlorogenic acid, and caffeine and the antioxidant capacities of coffee extracts [52,54,55]. The values obtained from the FRAP, DPPH, and Fe^2+^-chelating assays showed that the dGCB extract has greater antioxidant properties than other plant extracts used as natural antioxidants in meat preparation and meat products [42,48,56]. For this reason, the addition of small amounts into the burgers was considered sufficient to protect the meat matrix against peroxidation reactions, which can occur during refrigerated storage or cooking.

### 3.2. Total Phenolic Content (TCP), Antioxidant Activities, Heme (HI) and Non-Heme Iron (NHI) Content, and Lipid Oxidation of Raw and Cooked Burgers

Proximate analyses of the ground meats used to make burgers highlighted no significant differences between the two replicates (*p* > 0.05). For moisture, fat, crude protein, and ash contents (g/100 g), the mean values and the standard deviations were 67.1 ± 2.0, 11.8 ± 2.1, 20.5 ± 0.4, and 0.70 ± 0.05, respectively.

The total phenolic content, antioxidant capacities, and HI, NHI, and lipid oxidation values of raw and cooked burgers are reported in Table 2 and Table 3, respectively. The total phenolic content of the raw burgers was significantly affected by the formulation (Table 2).

Polyphenols were detected in two control burgers, although they were not added to these samples. The TPC value of the negative control (C) can be attributed to the phenolic groups of protein amino acids detected by the Folin–Ciocalteau assay. Similar TPC values were reported by Soriano et al. [7] in raw pork patties prepared without antioxidants. The further increase in the TPC value of the positive control (A) was due to the reaction of the ascorbate hydroxyl group in the Folin determination, as observed by Soriano et al. [7]. Comparing the negative control (C) with the burgers prepared by adding dGCB extract, the extra TPC measured in the P15, P30, and P60 burgers was in agreement with the amount of extract added and its TPC content (Table 1). In this study, the TPC data were not affected by storage time. On the other hand, Soriano et al. [7] reported a decrease in the TPC values of burgers prepared with ascorbic acid and natural antioxidants over the course of the storage time; this trend can be explained by the different packaging conditions applied in our study. The lack of oxygen protects the meat matrix against oxidation, preventing consumption of the antioxidant during storage. The TPC values of the cooked burgers were lower than the corresponding raw samples (Table 3). The value of the negative control (C) was similar to the value reported by Alarcón et al. [8] in a cooked pork model. The differences between the burgers prepared with different levels of dGCB extract and the negative control (C) were similar to those observed in raw meats, suggesting that the initial polyphenol quantity was not affected by heating.

In addition, the presence of polyphenol compounds may contribute to improvements in the cooked burgers’ nutritional profiles. The value likely increased over time due to the higher cooking loss observed after seven days of storage (Table 3). 

The antioxidant capacities of the raw and cooked burgers were measured by DPPH and FRAP assays, and the results are reported in Table 2 and Table 3, respectively. The first method measures the radical-scavenging activity, and the second method measures the ferric-reducing potential of a sample. Significant differences in antioxidant activities were observed between the formulations (Table 2). The meat antioxidant capacity of the burgers without an antioxidant (C) depends on the balance of and the interaction between endogenous anti and pro-oxidant substances present in the meat. Endogenous antioxidant systems include enzymes (i.e., superoxide dismutase, catalase, and glutathione peroxidase) and non-enzymatic hydrophilic and lipophilic compounds, such as vitamin C and E and ubiquinols, as well as proteins and peptides that are rich in histidine (e.g., carnosine and anserine) [41].

The raw burgers without an antioxidant (C) showed the lowest antioxidant activity; in these samples, the DPPH values were higher than the FRAP values (Table 2). Serpen et al. [41] observed that, in general, the reductive antioxidant power of raw meat was lower than the levels of DPPH-scavenging activity; they concluded that meat antioxidants likely have a low ability to reduce ferric iron to its ferrous form. The DPPH values of the negative control (C) were similar to data measured in pork burgers prepared by adding only 1.5% of salt and no antioxidant, as reported by Bellucci et al. [48]. Meanwhile, the FRAP values were in agreement with the results reported by Serpen et al. for pork meat [41]. 

As expected, the antioxidant activities of burgers were significantly increased by the addition of antioxidants such as ascorbic acid and dGCB extract. The radical-scavenging and ferric-reducing activities of P15 burgers were similar to those of the positive sample (A). A greater antioxidant capacity was observed in P30 burgers, and the highest value was measured in P60 burgers. Compared to the negative control (C), the ferric-reducing capacity increased more than the radical-scavenging capacity both in burgers prepared with ascorbic acid (A) and in those prepared with the dGCB extract (P15, P30, and P60). In addition, this difference grew larger between the P15 and P60 burgers. Likewise, Mancini et al. [57] observed a greater increase in ferric-reducing activity than in radical-scavenging activity in rabbit burgers prepared with ascorbic acid compared to control samples prepared without an antioxidant. The storage time appeared to not affect the radical-scavenging and ferric-reducing activities of all raw formulations. 

The same trends in antioxidant activity compared between formulations and storage times were observed in cooked pork burgers (Table 3). Cooking did not affect the radical-scavenging and ferric-reducing capacities of the negative control (C). Similar results were obtained by Serpen et al. [41] in the first minutes of heating minced pork meat in an oven at 180 °C. Comparing the cooked burgers containing antioxidants and the corresponding raw burgers, their ferric-reducing activity was more strongly affected by heating than the radical-scavenging activity, and the gap increased when moving from the positive control (A) and the lowest concentration of dGCB extract (P15) to burgers prepared with the medium and highest concentrations of the extract (P30 and P60). 

The HI and NHI contents were measured in raw and cooked burgers, and the results are shown in Table 2 and Table 3, respectively. The HI and NHI concentrations were not significantly different between the raw burger formulations (Table 2). The values are consistent with the results of pork meat cuts obtained by other authors [5,58,59]. The slightly higher values of NHI that were measured in the P30 and P60 burgers can probably be attributed to the presence of iron in the dGCB extract (Table 1). The HI and NHI concentrations did not reveal any differences between the storage times, even if a very slight decrease in HI and a simultaneous weak increase in NHI were observed. Zareian et al. [59] reported a 35% decrease in levels of heme iron and a corresponding increase in NHI content in pork meatballs packaged in a modified atmosphere after 12 days of storage at 4 °C. They observed that NHI content is highly correlated to myoglobin autoxidation, the denaturation rate, and the levels of metmyoglobin and ferryl myoglobin content. The vacuum packaging used in our study inhibited all mechanisms of myoglobin oxidation and degradation during the storage of raw burgers, limiting the release of iron; this could act as a pro-oxidant by enhancing lipid oxidation. 

Compared to their raw counterparts, the cooked burgers contained higher levels of HI and NHI (Table 3), due to moisture losses that occurred during cooking (Table 3). Similar differences between raw and cooked meat cuts were reported by Lombardi-Boccia et al. [58]. Many studies report that heating meat causes a partial loss of heme iron and the consequent growth of free iron [12,58]. The degree of heme destruction and iron release depends on the heating method and conditions (e.g., applied temperature, time). The cooking procedure used in this study slightly affected the ratio of HI to NHI. Likewise, Zhang et al. [12] reported a slight decrease in HI and increase in NHI in a control sample of pork tenderloin after being kept for 30 min at 80 °C in a water bath. 

Lipid oxidation in raw and cooked pork burgers was evaluated using a TBARS test and the results are reported in Table 2 and Table 3, respectively. The TBARS values of all raw burgers ranged between 0.16 and 0.25 mg MDA/kg, and differences were not found between the formulations and storage times. It is well known that lipid oxidation phenomena occur mainly by means of a reaction between unsaturated fatty acids and oxygen. Indeed, oxygen acts as a source of reactive oxygen species and is able to react with alkyl radicals to form peroxy radicals [21]. The vacuum packaging used in this investigation was able to protect the minced meat from lipid oxidation during storage.

The TBARS values of the cooked burgers were affected by the different formulations (Table 3). The negative control (C) had the highest values, increasing from 1.00 mg MDA eq./kg at T0 to 1.50 mg MDA eq./kg at T14; on the other hand, the positive control (A) showed lower TBARS values, ranging from 0.33 mg MDA eq./kg (T0) to 0.74 mg MDA eq./kg (T14). The pro-oxidant activities of salt and the efficacy of ascorbic acid in controlling lipid oxidation have been widely reported in the literature; similar trends were observed by Trujillo-Mayol et al. [60] in pan-grilled beef burgers and by Carballo et al. [61] in cooked lamb patties. On the contrary, Alarcón et al. [8] found higher TBARS values in cooked pork model systems prepared with sodium ascorbate (0.04%) than in the control without an antioxidant. The TBARS values of all cooked burgers prepared with the dGCB extract did not differ from the positive control (A), even at the lowest concentration (P15) tested in this study. However, the lipid oxidative stability was raised by doubling the extract concentration in the burgers, and the TBARS values of the P30 and P60 cooked burgers were much lower than 0.5 mg MDA eq./kg, which is considered the limit value for the sensory perception of the rancid flavor [48,56]. 

### 3.3. pH and Colorimetric Evaluation of Raw and Cooked Burgers during Storage

The pH and color parameters of raw pork burgers are reported in Table 4. The pH values did not differ significantly between the formulations and storage times. Likewise, Bellucci et al. [48] found that pH values were not affected by sodium erythorbate or by different concentrations of plant extract added to raw pork patties during the storage period. Other authors observed no changes in pH values when natural antioxidant extracts were added to pork burgers or beef patties, but the pH values of the samples decreased during refrigerated storage due to the growth of lactic acid bacteria [56,62]. The different trends found in our results during the storage period can be explained by the packaging conditions used in our study. 

The pH of the cooked burgers was not significantly influenced by the different formulations and storage time (Table 5). A similar pH trend was reported by Alarcón et al. [8] for cooked pork model systems prepared without antioxidants, with sodium ascorbate (0.04%), and with aqueous extracts (3%) of stems and vine shoots. Trujillo-Mayol et al. [60] showed the same results in pan-grilled beef burgers prepared without an antioxidant, with sodium ascorbate (0.1%), and with two different concentrations of avocado peel extract (0.5% and 1.0%) after 1 and 10 days of storage at 4 °C. 

The different formulations investigated did not affect the color parameters of raw pork burgers (Table 4). Bellucci et al. [48] reported no differences between samples without antioxidants and with sodium erythorbate, but all color parameter values decreased by adding açaí extract. Other authors noticed color changes in meat patties prepared with natural extracts rich in anthocyanins and flavonols [62]. On the other hand, Šojić et al. [56] found no significant differences in color parameters between the pork patties supplemented with two different extracts, namely terpenoid-rich Thymus serpyllum L. and the control, which had no added extracts. Similarly, at the concentrations tested in this study, the dGCB extract did not significantly affect the color parameters. Liu and Kitts [25] observed that the aqueous non-roasted coffee bean extract mainly contained chlorogenic acid and caffeine, and that its UV spectrum measured in the range of 250–700 nm was very similar to that of the colorless chlorogenic acid standard. The dGCB extract showed the same polyphenol composition as the extract prepared by Liu and Kitts [25], but the absorbance value, measured at 420 nm, was lower (Abs420 nm = 0.10) than that of their extract (Abs420 nm = 0.19). For all formulations, no discoloration occurred during storage (Table 4). The lack of oxygen prevented the oxidation of iron atoms in the heme group, further preventing the formation of the metmyoglobin responsible for the decrease in redness (a*) [11]. Moreover, the yellowness (b*), chroma, and hue were unaffected by storage time. The increase in lightness (L*) observed after seven days of storage for all treatments can be explained by a slight increase in moisture on the burgers’ surface. 

Color changes during cooking arise mainly because of the denaturation of globin and the oxidation of heme iron into its ferric form [16], but the color parameters of the cooked burgers were not affected by the different formulations (Table 5). Likewise, no color differences between samples without an antioxidant and with a synthetic antioxidant were reported by Alarcón et al. [8]. The lightness (L*) and the redness of all samples prepared with the dGCB extract were slightly lower than those of the positive control (A) and the negative (C) control, although there were no significant differences. Other authors found deep differences in the color of cooked meat products to which plant extracts were added [8].

The cooking loss values are reported in Table 3. This parameter was not affected by the formulation. Similar results were obtained by Bellucci et al. [48] for pork patties prepared with low concentrations of plant extracts. Jongberg at al. [63] reported that only high concentrations of phenolic compounds reduced the water-holding capacity of meat proteins due to the formation of large amounts of thiol–quinone adducts. After seven days of storage, the cooking loss significantly increased in all formulations (no F × ST interaction was observed). This result, as well as the trend in lightness (L*) in raw burgers (Table 4), suggests a probable loss in protein functionality over the storage period, regardless of the addition of an antioxidant.

### 3.4. Sensory Characteristics

The addition of natural antioxidants into meat and meat products requires the compliance of their sensory profile, especially when green coffee extract is used [38].

The sensory properties of raw burgers showed no significant differences in appearance, redness intensity, and color homogeneity (Figure 1a). The use of the coffee extract did not affect the presence of spots, proving that the dGCB extract is suitable for the meat matrix. Only in higher concentrations (P30 and P60) did the extract change the color of the raw burgers, leading to more intense brownness and the perception of a weak foreign aroma, without significant differences. This did not appear to reduce the overall acceptability of the raw burgers. This is in accordance with the results of burger patties reported by Soriano et al. [7], in which only higher concentrations of plant extracts partially modified the meat’s appearance and odor.

On the other hand, the color parameters of the raw burgers revealed that storage time reduced the scores with significant differences, particularly between T0 and T14 (Figure 1b). A lower red intensity value associated with a higher brown score negatively affected the burgers’ appearance and global acceptability values, as assessed by the panelists. Only the rancid odor was perceived to a significant degree at the end of the storage time, with a score that remained low; this result was also demonstrated by the TBARS values (Table 2), which did not impair the acceptability of the product.

The sensory characteristics of the cooked burgers are reported in Figure 2. The dGCB extract added at the highest levels influenced foreign aroma intensity (Figure 2a); the assigned scores remained low, slightly compromising the overall acceptability of the cooked burgers.

Elsewhere, the addition of green coffee extracts had no impact on the final sensory evaluation of fine-yeast pastry fried products, such as donuts [27]. Storage time partially affected overall acceptability, with lower values at the end of the storage period (Figure 2b), including increased scores for brown intensity and rancid odor, related to the lipid oxidation evidenced by TBARS tests; nevertheless, the sensory profile of the cooked burgers was not impaired.

## 4. Conclusions

The raw and cooked burgers prepared with 0.15% dGCB extract showed, throughout the shelf-life analysis, the antioxidant activity, lipid oxidation stability, and sensory profiles comparable to corresponding burgers prepared with 0.05% of synthetic ascorbic acid. In conclusion, dGCB extract is a promising natural antioxidant that could replace synthetic antioxidants, such as ascorbic acid, during the shelf life of vacuum-packed and refrigerated burgers. In addition, the use of this extract could minimize the number of agro-industrial by-products generated by the coffee industry.

## Figures and Tables

**Figure 1 foods-12-01264-f001:**
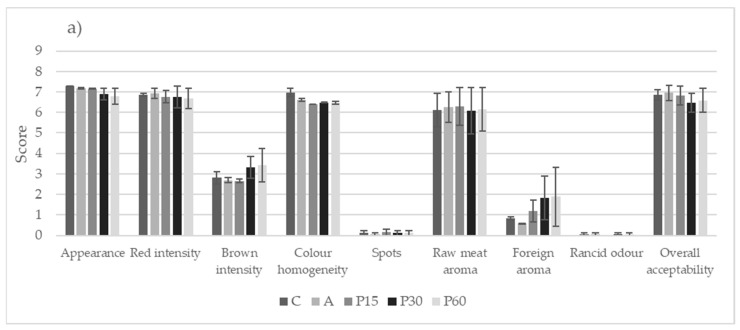
Sensory properties (appearance, red color intensity, brown color intensity, color homogeneity, presence of spots, intensity of raw meat aroma, presence of foreign aroma, intensity of rancid odor, and overall acceptability) of the raw burgers related to the formulation (**a**) and storage time (**b**). The results are expressed as estimated marginal means and standard error of the mean of each attribute, scored on a non-structured 0–9 intensity scale. Different letters associated with each sensory property indicate significant differences (Tukey’s test, *p* < 0.05). C = negative control, no antioxidant; A = positive control, +0.05% ascorbic acid; P15 = +0.15% dGCB extract; P30 = +0.30% dGCB extract; P60 = +0.60% dGCB extract. T0 = 0 days of refrigerated storage at 4 ± 1 °C; T7 = 7 days of refrigerated storage at 4 ± 1 °C; T14 = 14 days of refrigerated storage at 4 ± 1 °C.

**Figure 2 foods-12-01264-f002:**
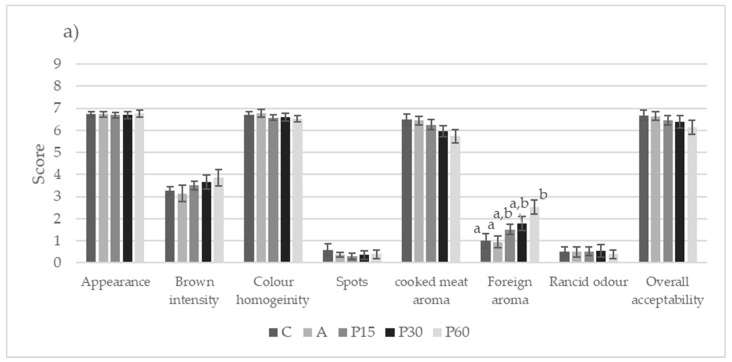
Sensory properties (appearance, brown color intensity, color homogeneity, presence of spots, intensity of cooked meat aroma, presence of foreign aroma, intensity of rancid odor, and overall acceptability) of the cooked burgers related to the formulation (**a**) and storage time (**b**). The results are expressed as estimated marginal means and standard error of the mean of each attribute, scored on a non-structured 0–9 intensity scale. Different letters assigned to each sensory property indicate significant differences (Tukey’s test, *p* < 0.05). C = negative control, no antioxidant; A = positive control, + 0.05% ascorbic acid; P15 = +0.15% dGCB extract; P30 = +0.30% dGCB extract; P60 = +0.60% dGCB extract. T0 = 0 days of refrigerated storage at 4 ± 1 °C; T7 = 7 days of refrigerated storage at 4 ± 1 °C; T14 = 14 days of refrigerated storage at 4 ± 1 °C.

**Table 1 foods-12-01264-t001:** Total phenolic content (TPC), chlorogenic acid, and caffeine concentration, antioxidant capacities (DPPH and FRAP), Fe^2+^-chelating activity, and total iron of the dGCB extract. The values are expressed as the mean ± standard deviation of three replicates.

Parameters	
TPC (g GA eq./kg)	114.4 ± 1.0
Chlorogenic acid (mg/g)	183.1 ± 19.3
Caffeine (mg/g)	246.4 ± 29.4
DPPH (mmol Trolox eq./kg)	960.3 ± 17.7
FRAP (mmol Trolox eq./kg)	1990.7 ± 63.1
Fe^2+^-chelating activity (mg/mL)	0.175 ± 0.001
Total iron (μg/g)	16.48 ± 0.84

**Table 2 foods-12-01264-t002:** Estimated marginal means and standard error of the mean (SEM) of total phenolic content (TPC, g GA eq. /kg), antioxidant capacities (DPPH and FRAP, mmol Trolox eq./kgdw), heme iron (HI, mg/100 g) and non-heme iron (NHI, mg/100 g), and lipid oxidation (TBARS; mg MDA eq./kg) of the raw pork burgers.

	Formulation ^1^							Storage Time ^2^				
	C	A	P15	P30	P60	SEM	*p*-Value	T0	T7	T14	SEM	*p*-Value
TPC	0.31 ^a^	0.59 ^b^	0.44 ^a.b^	0.59 ^b^	0.85 ^c^	0.05	***	0.55	0.54	0.59	0.04	NS
DPPH	13.46 ^a^	20.15 ^b^	24.84 ^b.c^	28.19 ^c^	34.05 ^d^	1.27	***	23.09	25.18	24.14	0.99	NS
FRAP	8.50 ^a^	19.65 ^b^	21.46 ^b^	33.38 ^c^	45.78 ^d^	1.51	***	24.98	27.15	25.14	1.17	NS
HI	0.55	0.57	0.55	0.55	0.56	0.02	NS	0.57	0.56	0.54	0.02	NS
NHI	0.16	0.16	0.16	0.17	0.17	0.02	NS	0.15	0.16	0.18	0.02	NS
TBARS	0.22	0.18	0.18	0.19	0.20	0.02	NS	0.19	0.18	0.21	0.02	NS

^1^: C = negative control, no antioxidant; A = positive control, + 0.05% ascorbic acid; P15 = +0.15% dGCB extract; P30 = +0.30% dGCB extract; P60 = +0.60% dGCB extract. Values with different letters indicate significant differences (*p*-value: NS = not significant; *** *p* < 0.001). ^2:^ T0 = 0 days of refrigerated storage at 4 ± 1 °C; T7 = 7 days of refrigerated storage at 4 ± 1 °C; T14 = 14 days of refrigerated storage at 4 ± 1 °C. Values with different letters indicate significant differences (*p*-value: NS= not significant; *** *p* < 0.001).

**Table 3 foods-12-01264-t003:** Estimated marginal means and standard error of the mean (SEM) of total phenolic content (TPC, g GA eq. /kg), antioxidant capacities (DPPH and FRAP, mmol Trolox eq./kgdw), heme iron (HI, mg/100 g) and non-heme iron (NHI, mg/100 g), lipid oxidation (TBARS; mg MDA eq./kg), and cooking loss percentage (CL%) of the cooked pork burgers.

	Formulation ^1^							Storage Time ^2^				
	C	A	P15	P30	P60	SEM	*p*-Value	T0	T7	T14	SEM	*p*-Value
TPC	0.24 ^a^	0.47 ^b^	0.37 ^a.b^	0.52 ^b^	0.75 ^c^	0.05	***	0.39 ^a^	0.49 ^a.b^	0.54 ^b^	0.04	*
DPPH	14.53 ^a^	19.75 ^a.b^	25.84 ^b.c^	27.57 ^b.c^	31.91 ^c^	2.49	***	23.44	25.14	23.18	1.76	NS
FRAP	7.49 ^a^	16.08 ^b^	18.64 ^b^	25.18 ^c^	36.00 ^d^	1.40	***	20.66	20.81	20.56	1.08	NS
HI	0.77	0.79	0.76	0.80	0.79	0.05	NS	0.78	0.79	0.78	0.04	NS
NHI	0.23	0.22	0.22	0.24	0.26	0.03	NS	0.23	0.23	0.25	0.02	NS
TBARS	1.28 ^b^	0.55 ^a.b^	0.47 ^a^	0.21 ^a^	0.20 ^a^	0.19	*	0.47	0.53	0.63	0.14	NS
CL%	17.3	18.4	18.3	16.9	17.1	2.66	NS	13.0 ^a^	19.7 ^b^	20.1 ^b^	2.06	*

^1^: C = negative control, no antioxidant; A = positive control, + 0.05% ascorbic acid; P15 = +0.15% dGCB extract; P30 = +0.30% dGCB extract; P60 = +0.60% dGCB extract. Values with different letters indicate significant differences (*p*-value: NS = not significant; * *p* < 0.05; *** *p* < 0.001). ^2:^ T0 = 0 days of refrigerated storage at 4 ± 1 °C; T7 = 7 days of refrigerated storage at 4 ± 1 °C; T14 = 14 days of refrigerated storage at 4 ± 1 °C. Values with different letters indicate significant differences (*p*-value: NS = not significant; * *p* < 0.05; *** *p* < 0.001).

**Table 4 foods-12-01264-t004:** Estimated marginal means and standard error of the mean (SEM) of the pH values and color parameters of the raw pork burgers.

	Formulation ^1^							Storage Time ^2^				
	C	A	P15	P30	P60	SEM	*p*-Value	T0	T7	T14	SEM	*p*-Value
pH	5.59	5.59	5.57	5.58	5.55	0.08	NS	5.60	5.56	5.58	0.06	NS
L*	52.9	53.4	52.6	52.5	52.1	0.68	NS	51.2 ^a^	53.0 ^b^	53.9 ^b^	0.53	**
a*	6.62	6.82	6.80	6.86	6.66	0.50	NS	6.97	6.84	6.45	0.39	NS
b*	12.2	12.2	12.6	12.7	12.9	0.48	NS	12.6	12.7	12.2	0.37	NS
Chroma	13.9	14.0	14.3	14.5	14.5	0.64	NS	14.4	14.5	13.8	0.50	NS
Hue	61.7	61.0	61.8	61.7	62.9	0.98	NS	61.3	61.9	62.2	0.76	NS

^1^: C = negative control, no antioxidant; A = positive control, + 0.05% ascorbic acid; P15 = +0.15% dGCB extract; P30 = +0.30% dGCB extract; P60 = +0.60% dGCB extract. Values with different letters indicate significant differences (*p*-value: NS = not significant; ** *p* < 0.01). ^2:^ T0 = 0 days of refrigerated storage at 4 ± 1 °C; T7 = 7 days of refrigerated storage at 4 ± 1 °C; T14 = 14 days of refrigerated storage at 4 ± 1 °C. Values with different letters indicate significant differences (*p*-value: NS = not significant; ** *p* < 0.01).

**Table 5 foods-12-01264-t005:** Estimated marginal means and standard error of the mean (SEM) of the pH, and color parameters of the cooked pork burgers.

	Formulation ^1^							Storage Time ^2^				
	C	A	P15	P30	P60	SEM	*p*-Value	T0	T7	T14	SEM	*p*-Value
pH	5.96	5.93	5.94	5.93	5.91	0.06	NS	5.99	5.90	5.91	0.04	NS
L*	60.5	60.0	58.5	58.8	57.8	0.88	NS	60.05	59.21	58.08	0.68	NS
a*	4.08	4.32	3.99	3.44	3.34	0.40	NS	3.79	3.71	3.99	0.31	NS
b*	14.2	14.5	14.4	13.9	13.9	0.04	NS	14.74	13.69	14.09	0.31	NS
Chroma	14.8	15.1	14.9	14.4	14.3	0.48	NS	15.24	14.20	14.66	0.37	NS
Hue	74.0	73.6	74.5	76.2	76.6	1.15	NS	75.62	74.95	74.37	0.89	NS

^1^: C = negative control, no antioxidant; A = positive control, + 0.05% ascorbic acid; P15 = +0.15% dGCB extract; P30 = +0.30% dGCB extract; P60 = +0.60% dGCB extract. Values with different letters indicate significant differences (*p*-value: NS = not significant). ^2:^ T0 = 0 days of refrigerated storage at 4 ± 1 °C; T7 = 7 days of refrigerated storage at 4 ± 1 °C; T14 = 14 days of refrigerated storage at 4 ± 1 °C. Values with different letters indicate significant differences (*p*-value: NS = not significant).

## Data Availability

Data are available from the corresponding author upon request.

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
