# Peer review of "Antioxidant and Sensory Properties of Raw and Cooked Pork Meat Burgers Formulated with Extract from Non-Compliant Green Coffee Beans"

_foods, 2023, doi:10.3390/foods12061264_

Round 1

Reviewer 1 Report

GENERAL CONSIDERATIONS:

The manuscript is very interesting, it brings up an important issue about the replacement of synthetic antioxidants in the context of the agri-food industry, combined with a sustainable proposal capable of increasing the quality of the food and its useful life, but it needs to be revised in some parts, as in the abstract, which does not contemplate the central objective of the study and the introduction, which does not bring the central problem of the study in a relevant way. The other sections also need to be reviewed on time.

SPECIFIC COMMENTS

TITLE: I have no suggestions.

HIGHLIGHTS: I have no suggestions.

ABSTRACT:

The abstract should contain more information about the work, mainly related to the purpose of the research. It should be inserted which best formulation was found throughout the research of those studied.

KEYWORDS: Add a word referring to the pecan by-product.

INTRODUCTION:

The theme is very relevant, but I suggest improving the approach to the importance and problems of the research. In addition, it does not bring the hypothesis of the study. In general, some paragraphs need to be revised in relation to the lack of references and improvement of the links between the themes addressed in the paragraphs.

Lines 31-37: I suggest that you approach your problem more precisely, inserting statistical data on the excessive consumption of sausages by the population, as in the case of hamburgers, their high perishability, which directly impacts the aspect of lipid oxidation and the color that is dealt with in the subsequent paragraph, addressing more about the problem of pork. Lines 35-37: Insert reference.

Lines 43-46: Insert reference.

Lines 55-59: I suggest readjusting the information in the text so that the link between paragraphs improves.

Lines 78-81: I suggest enriching your objective with the information that you want to identify whether such an extract can be used in place of the synthetic antioxidant, as you advocate this throughout your introduction.

MATERIALS AND METHODS:

Lines 239-239: Insert the methodology used for pH analysis and describe in more detail how it happened.

Lines 241-244: Insert the methodology used for analysis (Reference). Note: Were there previous tests on microbial control?

RESULTS AND DISCUSSION

Line 254-259: Were inconsistent patterns observed between treatments? This phrase is not a description of a result.

Was regression analysis performed? In which table are these results observed?

CONCLUSION

I suggest that the authors precisely indicate, based on the results found in the study, among the treatments observed, the best level of replacement and bring more general data on the main results of the research.

Author Response

Best regards

Monica Bergamaschi

Reviewer 2 Report

This study does not reflect its innovation in your manuscript, and there are confusing logical relationships in the full description, and the experimental design is not rigorous, lacking experimental indicators to support your conclusions, please review your manuscript and add experimental indicators again.

1. Abstract

1) Line 14: Please use the full name of “dGCB”, not the abbreviation.

2) Line 15: Please use the full name of “DPPH and FRAP”, not the abbreviation.

3) Line 21-23: The conclusion is vague, you should use mathematical methods statistical methods to explain your conclusions, do not touch the ambiguity.

4) Line 26-27: Can your experimental data results and indicators support your conclusions? Please study them carefully before.

2. Introduction

The preamble is logically confusing, it is recommended to reorganize and should highlight the main points of your research.

3. Material and methods Your research content is still meaningful, but the indicators used to support your conclusions are too few, it is recommended to increase the experimental indicators to produce experimental results, so as to more strongly support the conclusions of your research.

4. Results and discussion

1) Line 285: “The main polyphenol components observed in the dGCB extract were chlorogenic acid and caffeine”. This conclusion you get is very ambiguous, suggest to amend.

2) Line 302-303: It is recommended that instead of statements like 67.1 ± 2.0, 11.8 ± 2.1 in the results and analysis, you can just say the significance of the results and analysis.

3) Line 350: What "these compounds" refers to should be clearly stated.

Author Response

Best regards

Monica Bergamaschi

Reviewer 3 Report

the introduction should be decreased and focused more on the present study highlighting the relevance and significance of the work and discussing the gaps therein.

Table: Not clear 

The five treatments in the study were evaluated for their efficiency on 7 days interval but the values shown in table are just one which means only one sample was stored for storage study or only one was evaluated. But going through the material and methods section indicates that all five samples were evaluated. 

there is a clear mis interpretation/ analysis. Kindly check the table for exact analysis

Author Response

Best regards

Monica Bergamaschi

Round 2

Reviewer 1 Report

Thank you for your valuable manuscript but I really have one specific questions:

Please, remove in the Conclusion section: "The polyphenol-rich extract obtained from non-compliant defatted green coffee 585 beans (dGCB) by means of supercritical extraction was used as a natural antioxidant in 586 burgers."

You should not need to repeat your Objective in the Conclusion section.

Thank you.

Author Response

Dear Reviewer,

thank you for your further suggestions. Please see the attached letter

Reviewer 2 Report

The author has completed the paper in accordance with the revisions.

Author Response

Dear Reviewer,

thank you for your final approval. Please see the attached letter.

Best regards

Monica Bergamaschi

Reviewer 3 Report

The manuscript has been improved

Author Response

(The authors gave the same response as above.)
